# Regulation of RIPK1 Phosphorylation: Implications for Inflammation, Cell Death, and Therapeutic Interventions

**DOI:** 10.3390/biomedicines12071525

**Published:** 2024-07-09

**Authors:** Jingchun Du, Zhigao Wang

**Affiliations:** 1Department of Clinical Immunology, Kingmed School of Laboratory Medicine, Guangzhou Medical University, Guangzhou 510182, China; 2Center for Regenerative Medicine, Heart Institute, Department of Internal Medicine, Morsani College of Medicine, University of South Florida, 560 Channelside Drive, Tampa, FL 33602, USA

**Keywords:** RIPK1, inflammation, cell death, phosphorylation, dephosphorylation, kinase, phosphatase, apoptosis, necroptosis, therapeutics

## Abstract

Receptor-interacting protein kinase 1 (RIPK1) plays a crucial role in controlling inflammation and cell death. Its function is tightly controlled through post-translational modifications, enabling its dynamic switch between promoting cell survival and triggering cell death. Phosphorylation of RIPK1 at various sites serves as a critical mechanism for regulating its activity, exerting either activating or inhibitory effects. Perturbations in RIPK1 phosphorylation status have profound implications for the development of severe inflammatory diseases in humans. This review explores the intricate regulation of RIPK1 phosphorylation and dephosphorylation and highlights the potential of targeting RIPK1 phosphorylation as a promising therapeutic strategy for mitigating human diseases.

## 1. Introduction

Receptor-interacting protein kinase 1 (RIPK1) is a crucial regulator of inflammation and cell death. Recent findings indicate that both genetic mutations and non-genetic factors influencing RIPK1 activity can lead to a range of inflammatory and degenerative diseases, highlighting the necessity for precise regulation of RIPK1 function in maintaining human health [1,2].

The full-length human RIPK1 is a cytosolic protein consisting of 671 amino acids, with a molecular mass of approximately 76 kDa, sharing 68% identity with its mouse counterpart (Figure 1). Belonging to the RIP kinase family, RIPK1 is one of seven members, each featuring a homologous kinase domain (KD). Besides the common N-terminal kinase domain, RIPK1 possesses a C-terminal death domain (DD), facilitating its dimerization or interaction with other death-domain-containing proteins, such as TNFR1 (tumor necrosis factor receptor 1), TRADD (TNFR1-associated death domain protein), and FADD (Fas-associated death domain) [3]. Additionally, RIPK1 contains a bridging intermediate domain (ID) housing a RIP homotypic interaction motif (RHIM) [4]. The RHIM domain of RIPK1 facilitates its self-polymerization to form amyloid fibers [5]. It also allows interaction with other RHIM-containing proteins, such as RIPK3, ZBP1 (Z-DNA binding protein 1, also known as DAI and DLM-1), and TRIF (TIR-domain-containing adapter-inducing interferon β) [1,2].

RIPK1 primarily regulates inflammation through its scaffold function, while its involvement in cell death requires its kinase activity. The regulation of RIPK1 function involves various post-translational modifications, including ubiquitination, phosphorylation, and glycosylation. This review aims to summarize RIPK1’s impact on inflammation, cell survival, cell death, development, and disease pathogenesis, with a focus on the role of phosphorylation and dephosphorylation.

## 2. RIPK1-Mediated Pro-Survival and Inflammatory Signaling

RIPK1 was initially reported to strongly interact with the cell surface receptor Fas/APO-1 (CD95) and weakly with TNFR1 [6]. Subsequently, various members of the TNF superfamily, including TNFα, FasL, and TRAIL (TNF-related apoptosis-inducing ligand), were found to induce RIPK1-mediated NF-κB (nuclear factor kappa-light-chain-enhancer of activated B cells) activation [3]. Among these, the TNFα cascade has been extensively studied (Figure 2) [7]. Upon TNFα binding, RIPK1 and TRADD are rapidly recruited to TNFR1, initiating the assembly of complex I through mutual interactions between their death domains [8,9,10]. TRADD then recruits adaptor proteins TRAF2 and 5 (TNF receptor-associated factor protein 2/5), which in turn engage the E3 ubiquitin ligases cIAP1/2 (cellular inhibitor of apoptosis 1 and 2) [11,12]. cIAP1/2 catalyzes K63 ubiquitination of RIPK1, serving as a scaffold to recruit ubiquitin-binding proteins TAB2/3 (TAK1-binding protein 2/3) and TAK1 (transforming growth factor-β-activated kinase 1) [13,14]. TAK1 then activates the MAPK (mitogen-activated protein kinase) pathway, including p38 and MK2 (MAPK-activated kinase 2) [15]. Additionally, the K63 ubiquitin chain recruits another E3 complex LUBAC (the linear ubiquitin chain assembly complex), which catalyzes the M1 linear ubiquitin chains on RIPK1 and TNFR1 [16,17]. These linear ubiquitin chains recruit adaptor protein NEMO (NF-κB essential modulator), which further engages IKKα/β (IκB kinase α/β) and TBK1/IKKε [18,19,20]. IKKα/β subsequently activates the NF-κB pathway [21,22]. Both the MAPK pathway and the NF-κB pathway activate gene expression that promotes inflammation and cell survival while suppressing cell death (Figure 2) [7,15].

## 3. RIPK1-Mediated Apoptosis

Apoptosis is regarded as a non-inflammatory form of programmed cell death, during which the contents of the dying cells are contained within apoptotic bodies. Caspases, a subfamily of cysteine proteases are both the initiators and executioners of apoptosis, as revealed by genetic and biochemical studies. Physiological and pathogen-related stimuli can trigger apoptosis through either extrinsic or intrinsic pathways. This process is vital for maintaining normal development and tissue homeostasis [23,24].

RIPK1 primarily mediates extrinsic apoptosis induced by death receptor ligands, such as TNFα, FasL, and TRAIL, dependent on the cellular context [1]. Combining TNFα with a protein synthesis inhibitor cycloheximide (CHX), or an IAP antagonist Smac-mimetic, can switch the TNFα-induced inflammatory response to apoptosis (Figure 2) [25]. CHX promotes Caspase-8 activation by eliminating the endogenous Caspase-8 inhibitor c-FLIP (Cellular FLICE-inhibitory protein), leading to the formation of complex IIa, including TRADD, FADD, and Caspase-8 [26]. On the other hand, Smac-mimetic triggers cIAP1/2 auto-degradation, releasing RIPK1 from the complex I to form complex IIb, consisting of RIPK1, FADD, and Caspase-8 [25,27,28]. Activated Caspase-8 cleaves the preforms of caspase-3/7 to execute apoptotic death. Activation of TNFR1 under various deficient conditions (TAK1, NEMO, TBK1, IKKα/β, A20, and ABIN1) also triggers RIPK1 activation and complex IIb formation, leading to RIPK1-dependent apoptosis [29,30,31,32,33,34,35]. Importantly, RIPK1 kinase activity is essential for TNF and Smac-mimetic-stimulated, but not for TNF- and CHX-induced apoptosis, as demonstrated by RIPK1 knockdown, RIPK1 kinase inhibitor necrostatin, or kinase-dead mutants (K45A or D138N) [10,25,36,37,38].

## 4. RIPK1-Mediated Necroptosis

Necroptosis, a pro-inflammatory form of programmed cell death, is characterized by cell membrane rupture and the release of damage-associated molecular patterns (DMAPs). While not essential for embryogenesis, necroptosis plays a vital role in immune defense against pathogens and is implicated in various human diseases, including inflammation, tissue damage, and neurodegeneration [39].

Necroptosis is initially suppressed by apoptosis, primarily through the cleavage of RIPK1 by activated Caspase-8 in complex II, for example, when induced by TNF and Smac-mimetic or CHX [40,41]. However, when Caspase-8 is inactivated by specific inhibitors (such as Z-VAD-FMK) or genetic elimination, activated RIPK1 in complex IIb recruits RIPK3 through their respective RHIM domains, initiating the formation of another protein complex, called the necrosome (Figure 2) [5,42,43,44,45,46]. Oligomerized RIPK3 then recruits the casein kinase 1 family proteins CK1α/δ/ε, which phosphorylate Ser227 of human RIPK3 [47]. Phosphorylated RIPK3 subsequently recruits MLKL and phosphorylates human MLKL at Thr357 and Ser358 [48,49]. Consequently, MLKL undergoes oligomerization into tetramers and amyloid-like polymers, which translocate to the plasma membrane, resulting in plasma membrane permeabilization [50,51,52,53,54,55,56]. In addition, activated MLKL translocates to the lysosomal membrane, where it forms amyloid-like polymers to facilitate lysosomal membrane permeabilization and the release of lysosomal proteases, thereby promoting cell death [57].

RIPK1-mediated necroptosis requires its kinase activity, similar to its involvement in RIPK1-dependent apoptosis. For example, the RIPK1 inhibitor Nec-1 effectively prevents necroptosis induced by TNF, TLR ligands, and interferons [43,58,59]. Moreover, mice with kinase-dead RIPK1 knock-in mutations are resistant to TNF-induced necroptosis and systemic inflammatory response syndrome (SIRS), similar to RIPK3 knockout mice, and demonstrate superior resistance compared to MLKL knockout mice [38,60,61].

Multiple innate immune signaling molecules, including death receptors (such as TNFR1), pathogen recognition receptors (such as Toll-like Receptor TLR3 and TLR4), and the cytosolic RNA sensor ZBP1, can induce necroptosis. Activation of these pathways leads to the interaction between the RHIM domains of proteins, such as RIPK1, TRIF, or ZBP1, with the RHIM domain of RIPK3, activating RIPK3 and MLKL to promote necroptosis [58,62,63,64].

The role of apoptosis in suppressing necroptosis is crucial for embryonic development. Deficiency in apoptosis components, such as the knockout of Caspase-8 or FADD, often results in late-gestation embryonic lethality, primarily due to hyperactivation of necroptosis [65,66]. Simultaneous deletion of RIPK3 or MLKL can rescue the embryonic lethality of these mice, albeit with immune deficiencies in adulthood [67,68,69].

## 5. RIPK1-Mediated Pyroptosis and PANoptosis

Pyroptosis is another form of immunogenic programmed necrosis, characterized by the activation of inflammatory caspases, such as Caspase-1, 4, 5, and 11. These activated caspases cleave gasdermin family proteins to release their N-terminal pore-forming domain and trigger cell death [23,24,70,71]. Pyroptosis plays a critical role in innate defense against pathogens by eliminating infected host cells, thereby removing the breeding ground for pathogens, and activating the inflammatory response for pathogen clearance. There is extensive crosstalk among programmed cell death pathways. When pyroptosis, apoptosis, and necroptosis occur simultaneously, such as under pathogen infections, the combination of these cell deaths is defined as PANoptosis. Concurrent activation of all three cell death pathways enables the evasion of pathogen-mediated inhibition of individual pathways, thereby enhancing host defense [72].

Recent findings have uncovered the role of RIPK1 in regulating pyroptosis and PANoptosis. For example, in the Gram-negative bacteria *Yersinia* infection, RIPK1-dependent activation of Caspase-8 cleaves gasdermin D and E, inducing pyroptosis in mouse macrophages [73,74,75,76]. Furthermore, *Yersinia* infection also modulates RIPK1-dependent apoptosis and necroptosis, concurrently with pyroptosis activation, thus triggering PANoptosis [77,78].

## 6. Phosphorylation of RIPK1

### 6.1. Auto-Activating Phosphorylation

The serine–threonine kinase activity of RIPK1 is crucial for both complex IIb-dependent apoptosis and necroptosis [79]. Typically, kinases adopt a closed conformation and require phosphorylation in the activation loop, also known as the T-loop, to activate their kinase activity [80]. These activating phosphorylation events can be catalyzed by upstream kinases or achieved through autophosphorylation. Currently, autophosphorylation is the only known mechanism for activating RIPK1 (Figure 3). For instance, autophosphorylation of S161 stabilizes the open conformation of the T-loop and promotes human RIPK1 kinase activation to induce necroptosis [43]. Furthermore, mitochondrial reactive oxygen species (ROS) modify three essential cysteine residues of RIPK1, leading to cysteine-mediated aggregation of RIPK1 and subsequent autophosphorylation on S161, which is critical for RIPK1 to effectively promote necrosome formation and cell death [81]. Moreover, S166 autophosphorylation of RIPK1 is indispensable for MLKL activation and necrosome formation. Mutation of S166 effectively prevents multiple RIPK1 kinase-dependent inflammatory lesions in vivo, such as intestinal colitis, hepatitis, liver tumorigenesis, skin inflammation, and TNF-induced SIRS. Interestingly, while autophosphorylation of Ser166 is essential, it alone is not adequate to initiate RIPK1-mediated cell death [82]. Multiple autophosphorylation sites, including serine residues 14/15, 20, 161, and 166, cooperate to induce conformational changes in RIPK1 [43,83]. These changes facilitate its association with cell death effectors, such as FADD and RIPK3, promoting the assembly of cell-death-inducing signaling complexes, such as complex II and the necrosome. It is noteworthy that the recombinant RHIM domain of RIPK1 exhibits a significantly higher affinity toward itself than the RHIM domain of RIPK3 [5]. Autophosphorylation of RIPK1 is thought to change its conformation, favoring the interaction between the RIPK1 and RIPK3 RHIM domains over the interactions between RIPK1 RHIM domains, thereby promoting necrosome formation [47,84].

### 6.2. Inhibitory Phosphorylation

The kinase activity of RIPK1 is tightly controlled at multiple levels to prevent spontaneous activation. Various post-translational modifications on RIPK1, such as ubiquitination and inhibitory phosphorylation, are intricately connected to keep RIPK1 kinase activity in check. For instance, in complex I, RIPK1 undergoes K63 ubiquitination by cIAP1/2 and M1 linear ubiquitination by LUBAC. These modifications stabilize complex I, inhibiting its dissociation and formation of cell-death-promoting complex II. In addition to activating the MAPK pathway and the NF-κB pathway to activate gene expression that promotes cell survival and inflammation, TAK1 and IKK kinases further suppress cell death by performing inhibitory phosphorylation on RIPK1 to block its kinase activity (Figure 3). For instance, TAK1 activates MK2, which directly phosphorylates S320 and S335 of human RIPK1, or S321 and S336 of mouse RIPK1, to inhibit RIPK1 kinase activity and subsequent apoptosis or necroptosis [85,86,87]. Interestingly, TAK1 is also reported to directly phosphorylate mouse RIPK1 at S321 [29]. In addition, TAK1 activates IKKα/β, which in turn phosphorylates S25 in the kinase domain of RIPK1. Phosphorylation of S25 prevents ATP binding and inhibits RIPK1 kinase activation [88]. Furthermore, TBK1/IKKε phosphorylates T189 in the kinase domain to inhibit RIPK1 kinase activity [31,32]. It is important to note that MK2 phosphorylates cytosolic RIPK1, while IKKα/β, TBK1/IKKε, and TAK1 phosphorylate ubiquitinated RIPK1 in complex I.

Recently, kinases outside of the TNF pathway have also been found to directly phosphorylate RIPK1 to inhibit its kinase activity and cell death. For example, glucose starvation activates AMPK (adenosine monophosphate-activated protein kinase), which phosphorylates S416 of human RIPK1 (or S415 of mouse RIPK1) to inhibit RIPK1 kinase activity and cell death [89].

In addition to serine/threonine phosphorylation, tyrosine phosphorylation has also been found to inhibit RIPK1 activity. Studies have shown that JAK1 (Janus Kinase 1) and Src kinases phosphorylate Y384 of human RIPK1 (or Y383 of mouse RIPK1) to inhibit RIPK1 kinase activity and subsequent cell death [90].

Inhibitory phosphorylation of RIPK1 plays a pivotal role in host defense against pathogens and modulates inflammatory responses. For example, the Gram-negative bacterial pathogen *Yersinia* counters the host defense by inhibiting NF-κB- and MAPK-mediated pro-inflammatory cytokines’ expression, while promoting RIPK1 activation-dependent cell death [91]. Its effector protein, acetyltransferase YopJ, elicits multiple functions in the process. First, it inactivates IKKα/β and TAK1 to block NF-κB and MAPK activation [92]. Second, it blocks the inhibitory phosphorylation of S25 of RIPK1 by IKKα/β to promote RIPK1-dependent macrophage cell death [88]. Lastly, it inactivates MK2, preventing inhibitory phosphorylation of S321 and S335 on RIPK1, thereby activating RIPK1-dependent cell death [85,86,87]. As a consequence, mice expressing the S25D-RIPK1 mutant fail to activate RIPK1-dependent cell death and are defective in defending against *Yersinia* infection, similar to the mice expressing the RIPK1 kinase-dead mutant K45A [88]. In addition, inhibitors of TAK1, IKKα/β, IKKε, and MK2, the kinases responsible for the inhibitory phosphorylation of RIPK1, all exacerbate TNF-induced necroptosis and SIRS [29,31,33,86].

Together, the inhibitory phosphorylation events by these kinases function as crucial checkpoints to prevent RIPK1 kinase activation and subsequent cell death. Dysregulation of any of these inhibitory phosphorylation events leads to elevated cell death and is frequently associated with inflammatory diseases.

## 7. Dephosphorylation of RIPK1

Protein phosphatases play a complementary role in regulating phosphorylation homeostasis. These enzymes are classified into three main families based on the sequence similarity of the catalytic domain and substrate specificity: PTPs (protein tyrosine phosphatases), PPPs (phosphoprotein phosphatases), and PPMs (protein phosphatase metal-dependent). PTPs specifically dephosphorylate phospho-tyrosine residues, while PPPs and PPMs dephosphorylate phospho-serine and phospho-threonine residues. In addition, a subfamily of PTPs, called the dual-specificity phosphatases, dephosphorylate all three phospho-amino acids. PPPs and PPMs differ in that PPMs require metal ions, such as magnesium or manganese, for their activity and function as single-subunit enzymes, while PPPs require regulatory subunits [93].

PP1 (protein phosphatase 1) is an important subfamily of PPPs. Its catalytic subunits (PP1c), including PP1α, PP1β, and PP1γ, are responsible for dephosphorylation of the majority phospho-serine and phospho-threonine sites in mammalian cells, regulating a broad range of cellular processes. Each PP1 catalytic subunit is obligatorily complexed with one or two regulatory subunits to form distinct PP1 holoenzymes. The regulatory subunits, also known as PP1-interacting proteins (PIPs) or regulatory interactors of protein phosphatase one (RIPPOs), determine substrate specificity by directing PP1c to the subcellular locations of its substrates and modulating its activity toward different substrates. There are approximately 200 validated PIPs, which assemble into more than 650 different PP1 holoenzymes in mammalian cells, enabling the dephosphorylation of diverse substrates [94].

While numerous kinases have been identified to phosphorylate RIPK1, only a limited number of phosphatases are found to dephosphorylate RIPK1 or RIPK3. For example, Ppm1b, a metal-ion-dependent phosphatase, dephosphorylates and inactivates RIPK3 to prevent the recruitment of MLKL into the necrosome, thus inhibiting subsequent necroptosis. Moreover, *Ppm1b^−/−^* mice exhibited heightened sensitivity to TNF-induced SIRS compared to WT mice, confirming its role in inhibiting necroptosis in vivo [95].

A sensitized CRISPR whole-genome knockout screen revealed that PPP1R3G (protein phosphatase 1 regulator subunit 3G) is essential for necroptosis [96]. Specifically, PPP1R3G forms a holoenzyme with PP1γ to directly dephosphorylate the inhibitory phosphorylation sites of human RIPK1, including S25, S320, and S335, thereby activating RIPK1-dependent apoptosis and necroptosis (Figure 4). An interesting note is that the holoenzyme does not remove the activating phosphorylation of S166 in vitro. In this context, upon treatment with TNF/Smac-mimetic/Z-VAD-FMK (T/S/Z), TRAF2 interacts with PPP1R3G to recruit the PPP1R3G/PP1γ holoenzyme to complex I, where PP1γ dephosphorylates the inhibitory phosphorylation sites of RIPK1, activating RIPK1 kinase. Loss of PPP1R3G leads to loss of RIPK1 autophosphorylation at S166 and subsequent failure to form complex IIb to induce cell death. Like many other PP1 regulatory subunits, PPP1R3G interacts with PP1γ through an RVXF motif (X stands for any amino acids) [97]. Mutation of RVQF in PPP1R3G to RAQA disrupts the interaction with PP1γ. Importantly, the RAQA mutant fails to rescue RIPK1 activation and cell death in PPP1R3G knockout cells. Furthermore, prevention of RIPK1 inhibitory phosphorylation with p38 or IKK inhibitors or mutation of serine 25 of RIPK1 to alanine largely restores cell death in PPP1R3G-knockout cells. Finally, *Ppp1r3g^−/−^* mice are protected from TNF-induced SIRS, confirming the important role of PPP1R3G in regulating apoptosis and necroptosis in vivo. Due to experimental sensitivity limitations, the authors were unable to determine if PPP1R3G/PP1γ removes the inhibitory phosphorylation of T189. This warrants further analysis in the future. Additionally, it will be interesting to investigate if the PPP1R3G/PP1γ holoenzyme removes inhibitory phosphorylation of S415.

A recent study revealed that the PPP6C (protein phosphatase 6 catalytic subunit) is essential for necroptosis, identified through a CRISPR whole-genome knockout screen [98]. As previously reported [99], PPP6C is recruited to complex I through TAB2, and dephosphorylates TAK1 to prevent inhibitory phosphorylation of RIPK1, thus activating TNF-induced necroptosis. Gastrointestinal tract-specific deletion of one allele of *Ppp6c* in mice could partially alleviate cecum damage caused by TNF-induced SIRS, confirming its role in necroptosis activation. Recent findings have partially corroborated this result, as evidenced by a CRISPR screen that identified the protein phosphatase 6 (PP6) holoenzyme as an activator of TAK1 inhibitor-induced PANoptosis [100]. The study demonstrated that ablation of the catalytic subunit PPP6C, or combined deletion of its regulatory subunits PPP6R1, PPP6R2, and PPP6R3, resulted in enhanced inhibitory phosphorylation of RIPK1 at S321 and reduced auto-activating phosphorylation at S166, consequently suppressing PANoptosis. However, the research did not conclusively determine whether the PP6 complex directly dephosphorylates RIPK1 at S321 or if it acts on TAK1, as proposed in the previous study [98]. This leaves open questions regarding the precise mechanism of PP6 in regulating RIPK1 phosphorylation.

## 8. RIPK1 in Development

The scaffolding function, rather than kinase activity, of RIPK1, plays an important pro-survival role in regulating early postnatal lethality and inflammatory response by preventing apoptosis and necroptosis. Specifically, the death domain of RIPK1 binds the death domain of FADD to prevent FADD and Caspase-8-dependent apoptosis, while the RHIM domain of RIPK1 binds RHIM domains of RIPK3 and ZBP1, preventing their hyperactivation-induced necroptosis. For example, genetic deletion of RIPK1 in mice causes postnatal lethality [101]. While double-knockout of RIPK3, Caspase-8, or FADD, along with RIPK1, only marginally prolongs survival [102,103,104], triple-knockout of RIPK1, RIPK3, and either Caspase-8 or FADD rescues RIPK1-deficient mice, allowing them to survive weaning and mature normally [105,106]. The RHIM domain of RIPK1 inhibits ZBP1–RIPK3–MLKL-mediated necroptosis, crucial for preventing late embryonic lethality and adult skin inflammation [107,108]. Moreover, RIPK1 is essential for maintaining the survival of intestinal epithelial cells (IECs) by blocking apoptosis and necroptosis [109]. Additionally, mice harboring RIPK1 kinase-dead knock-in mutants, including D138N and K45A, survive to adulthood with no gross or histological abnormalities, indicating that RIPK1 kinase activity is dispensable for survival [37,38].

## 9. RIPK1-Mediated Inflammatory Diseases

Many human inflammatory and neurodegenerative diseases are associated with abnormal RIPK1 expression or activity. Reports of gene mutations or non-genetic factors that affect RIPK1 activity are accumulating, highlighting the importance of RIPK1 regulation in human diseases.

Reduced RIPK1 expression can lead to various human diseases, largely due to the hyperactivation of RIPK3, ZBP1, and Caspase-8. As discussed previously, RIPK1 neutralizes RIPK3 and ZBP1 through RHIM domain interaction under normal conditions, and loss of RIPK1 leads to overactivation of ZBP1 and RIPK3, resulting in excessive necroptosis and systemic inflammation. In the meantime, RIPK1 inhibits FADD/Caspase-8-mediated apoptosis through death domain interaction during development, and loss of RIPK1 leads to excessive apoptosis. In humans, rare homozygous loss-of-function (LoF) mutations in RIPK1, including missense, nonsense, and frameshift mutations, cause combined immunodeficiency and inflammatory bowel disease (IBD). Many of these patients also suffer from lymphopenia, recurrent infections, and arthritis [110,111,112,113].

Conversely, elevated RIPK1 activity is also implicated in various human diseases due to heightened inflammation and cell death. For instance, rare mutations in RIPK1, such as D324N, D324H, and D324Y at the Caspase-8 recognition site LQLD, block Caspase-8-mediated cleavage of RIPK1, resulting in an autosomal-dominant autoinflammatory disease, characterized by recurrent fevers and lymphadenopathy [114,115,116]. Patients with these variants often have increased pro-inflammatory cytokines and chemokines, such as IL-6, TNF, and CXCL2/3, and their peripheral blood mononuclear cells are hypersensitive to RIPK1 activation-dependent apoptosis and necroptosis induced by TNF.

Furthermore, mutations in other genes that result in hyperactivation of RIPK1 kinase activity also lead to human diseases. For instance, monogenic mutations in genes such as *IKBKG* (encoding NEMO), *TNIP1* (encoding ABIN1), *TNFAIP3* (encoding A20), and members of the *LUBAC* complex have been linked to auto-immune and inflammatory disorders, such as inflammatory bowel disease, psoriasis, rheumatoid arthritis, and multiple sclerosis [117,118,119,120]. Interestingly, these genes are also involved in regulating NF-κB signaling [121]. Animal model studies have demonstrated that genetic or pharmacological inhibition of RIPK1 kinase activity can alleviate pathological symptoms, indicating that the pathogenesis resulting from these mutations may be driven more by dysregulated RIPK1-dependent cell death rather than a failure to activate NF-κB [35,60].

Several chronic neurodegenerative diseases, such as amyotrophic lateral sclerosis (ALS), Alzheimer’s disease (AD), and Parkinson’s disease (PD), are also linked to increased activation of RIPK1 [122,123,124]. For example, mutations in the *optineurin* (*OPTN*) gene have been implicated in human ALS. In mouse models, loss of *OPTN* leads to elevated RIPK1 activity, as well as downstream RIPK3 and MLKL activation, resulting in axon degeneration, which is partially rescued by *Ripk3* loss-of-function or treatment with a RIPK1 inhibitor necrostatin [122]. Furthermore, an aging-induced reduction in TAK1 expression combined with *TBK1* mutations promotes the onset of neurodegenerative diseases, including ALS and frontotemporal dementia (FTD). This is mainly attributed to the hyperactivation of RIPK1, due to decreased inhibitory phosphorylation resulting from reduced activity of TAK1 and TBK1. Importantly, the ALS/FTD phenotype is partially rescued by a single allele of kinase-dead RIPK1 [32].

## 10. Therapeutic Perspectives

Elevated RIPK1 activity is associated with numerous human diseases, making it a crucial target for therapeutic interventions. In theory, RIPK1 kinase inhibitors will prevent RIPK1 hyperactivation-induced inflammatory diseases, while preserving RIPK1 scaffold function to maintain its basal inhibition on RIPK3, ZBP1, and FADD/Caspase-8, thus averting unwanted necroptosis or apoptosis. Indeed, in mouse models, RIPK1 inhibitors have been shown to prevent or alleviate clinical symptoms of various diseases, including SIRS, ischemia-induced tissue injury, neurodegeneration, and bacterial and viral infections [2]. Currently, numerous RIPK1 inhibitors are in different phases of clinical trials for a spectrum of human inflammatory diseases, ranging from rheumatoid arthritis, cutaneous lupus erythematosus, ulcerative colitis, SARS-CoV-2 infection, to Alzheimer’s disease and ALS (Table 1) [125,126,127,128,129,130,131] (ClinicalTrials.gov). Many of these RIPK1 inhibitors have successfully passed phase I safety tests, revealing important dose-dependent effects. For example, the brain-penetrant RIPK1 inhibitor, DNL104, demonstrated a clear dose-response relationship in its safety profile. In the single-ascending-dose group, DNL104 was well tolerated across a range of doses, indicating a favorable safety profile at lower concentrations. However, in the multiple-ascending-dose group, 37.5% of subjects experienced post-dose liver toxicity, highlighting the potential for adverse effects at higher cumulative doses or with prolonged exposure [125]. This dose-dependent toxicity underscores the importance of careful dose optimization in RIPK1 inhibitor development. Some inhibitors have progressed to phase II trials, where dose-ranging studies are further refining the therapeutic window and optimal dosing regimens. However, as yet, no RIPK1 inhibitors have advanced to phase III trials, partly due to the ongoing process of establishing the most effective and safe dosing strategies for long-term treatment.

Considering the diverse functions of RIPK1 and the uncertain outcomes of the clinical trials involving RIPK1 inhibitors, there is a pressing need to identify novel targets for specifically inhibiting its cell-death-promoting activity. Due to the pivotal role of PPP1R3G/PP1γ in removing inhibitory phosphorylation sites on RIPK1, it emerges as a promising alternative therapeutic target. Notably, PPP1R3G interacts with PP1γ through a short RVQF motif, presenting a unique opportunity to develop short-peptide mimetics that disrupt PPP1R3G and PP1γ interaction. This disruption could potentially block RIPK1-dependent apoptosis and necroptosis. The same approach has been successfully employed in designing the Smac-mimetics, which mimics the four-residue AVPI sequence in the SMAC protein. These mimetics specifically mimic the interaction between SMAC and IAPs to induce IAP degradation, thereby activating apoptosis [27,28,132]. Unlike RIPK1 inhibitors, these inhibitors of PPP1R3G/PP1γ maintain the inhibitory phosphorylation sites on RIPK1, preventing its hyperactivation, while not altering RIPK1 scaffold function, thereby preserving its other functions. At the same time, these inhibitors would have minimal impact on the phosphatase activity of PP1γ, thus maintaining its other vital functions. This innovative approach holds potential for therapeutic interventions targeting inflammatory diseases associated with heightened RIPK1 activity, while minimizing any adverse effects on cellular homeostasis.

## 11. Conclusions and Future Perspectives

The pivotal role of RIPK1 in orchestrating cell survival and cell death necessitates precise regulation of its activity. While phosphorylation is a key regulatory mechanism, other post-translational modifications, such as ubiquitination and glycosylation, also profoundly influence RIPK1’s scaffolding function and kinase activity. However, the complex interplay among these modifications and their collective impact on RIPK1 function remain to be fully deciphered. Future research should focus on elucidating the temporal and spatial dynamics of these modifications, their cross-talk, and their responses to varying cellular contexts. Unraveling this intricate regulatory network holds promise for developing more targeted and effective treatments for RIPK1-related conditions. By identifying novel intervention points, researchers may achieve fine-tuned control over cell death and inflammatory processes in various pathological settings. Moreover, exploring the potential of combining RIPK1-targeted therapies with other treatments, particularly in cancer and inflammatory diseases, represents an exciting avenue for future investigation. This approach could lead to more comprehensive and efficacious therapeutic strategies, potentially revolutionizing the treatment of these complex disorders.

## Figures and Tables

**Figure 1 biomedicines-12-01525-f001:**
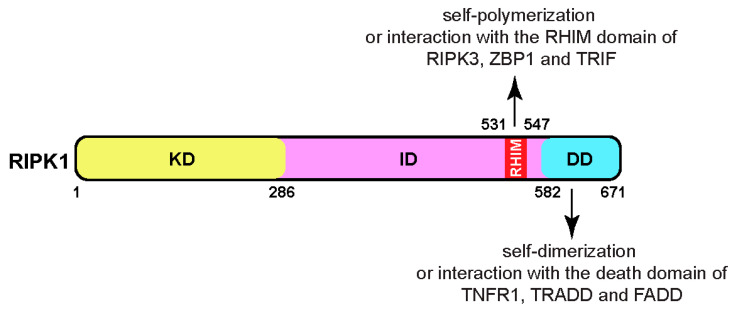
The domain structure of RIPK1 comprises a kinase domain (KD), an intermediate domain (ID), and a death domain (DD). Within the intermediate domain lies the RIP homotypic interaction motif domain (RHIM), which participates in polymerization and interacts with the RHIM domains of RIPK3, ZBP1, and TRIF. The death domain of RIPK1 facilitates homo-dimerization and interacts with the death domains of TNFR1, TRADD, and FADD. Abbreviations: RIPK1, receptor-interacting protein kinase 1; RIPK3, receptor-interacting protein kinase 3; ZBP1, Z-DNA-binding protein 1, also known as DAI (DNA-dependent activator of interferon regulatory factors) and DLM-1; TRIF, TIR-domain-containing adapter-inducing interferon β; TNFR1, tumor necrosis factor receptor 1; TRADD, TNFR1-associated death domain protein; FADD, Fas-associated death domain.

**Figure 2 biomedicines-12-01525-f002:**
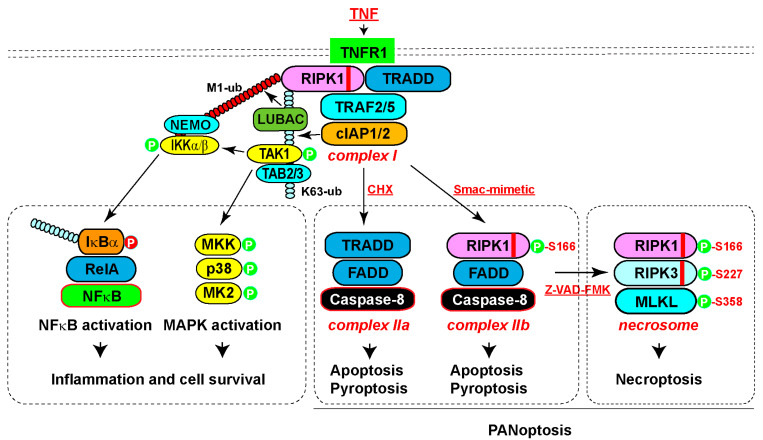
Pleiotropic TNF signaling pathways. (1) Inflammation and cell survival. Engagement of TNF with its receptor TNFR1 leads to the recruitment of RIPK1 and TRADD through death domain interactions to initiate complex I formation. TRADD then recruits adaptor protein TRAF2/5, which binds E3 ligase cIAP1/2. cIAP1/2 catalyzes K63 ubiquitination of RIPK1, serving as a scaffold to recruit ubiquitin-binding proteins TAB2/3 and associated TAK1, activating the downstream MAPK pathway. Additionally, the K63 ubiquitin chain recruits another E3 complex, LUBAC, which catalyzes the M1 linear ubiquitin chains on RIPK1 and TNFR1. These linear ubiquitin chains recruit adaptor protein NEMO and associated IKKα/β, phosphorylating IκBα to promote its degradation and subsequent NF-κB activation. Both the MAPK pathway and the NF-κB pathway activate gene expression, which promotes cell survival and inflammation. (2) Apoptosis and/or pyroptosis. Under TNF treatment with protein synthesis inhibition by cycloheximide (CHX), complex I is converted to complex IIa, containing TRADD, FADD, and Caspase-8, leading to oligomerization and activation of Caspase-8 and subsequent apoptosis. Alternatively, co-treatment of TNF with a cIAP1/2 inhibitor, Smac-mimetic, converts complex I to complex IIb, containing RIPK1, FADD, and Caspase-8, which also activates Caspase-8 and apoptosis. Under some circumstances, such as during *Yersinia* infection, activated Caspase-8 cleaves gasdemin D or E to trigger pyroptosis. (3) Necroptosis. Inhibition of apoptosis with Z-VAD-FMK, along with the presence of RIPK3, leads to the conversion of complex II into the necrosome. The core components of the necrosome include RIPK1, RIPK3, and MLKL, resulting in polymerization and membrane translocation of MLKL and subsequent cell death. In general, the scaffold function of RIPK1 is important for inflammation and cell survival, while the kinase activity is important for complex IIb-dependent apoptosis as well as necroptosis. Under some circumstances, such as during pathogen infection, simultaneous activation of pyroptosis, apoptosis, and necroptosis occurs, which is defined as PANoptosis. Abbreviations: TNF, tumor necrosis factor; TRAF2/5, TNF receptor-associated factor protein 2/5; cIAP1/2, cellular inhibitor of apoptosis 1 and 2; TAB2/3, TAK1-binding protein 2/3; TAK1, transforming growth factor-β-activated kinase 1; LUBAC, the linear ubiquitin chain assembly complex; NEMO, NF-κB essential modulator; IKKα/β, IκB kinase α/β; IκBα, inhibitor of kB alpha; NF-κB, nuclear factor kappa-light-chain-enhancer of activated B cells; MAPK, mitogen-activated protein kinase; MLKL, mixed-lineage kinase-like protein; CHX, cycloheximide; Smac-mimetic, Second Mitochondria-derived Activator of Caspases-mimetic. The red stripes in the diagram of RIPK1 and RIPK3 indicate the RHIM domain.

**Figure 3 biomedicines-12-01525-f003:**
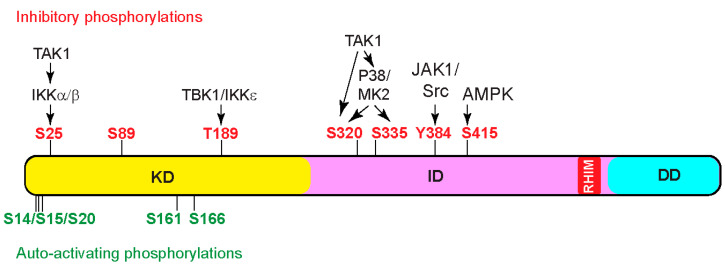
Phosphorylation events impacting RIPK1 kinase activity. Schematic representation of RIPK1, showing key phosphorylation sites. Green characters denote auto-activating phosphorylation events, while red characters denote inhibitory phosphorylation events. Multiple kinases catalyze inhibitory phosphorylation events, which serve as critical checkpoints for cell death activation. Notably, the kinase responsible for S89 phosphorylation has not yet been reported. Abbreviations: TBK1, TANK-binding kinase 1; IKKε, IκB kinase ε; MK2, MAPK-activated protein kinase 2; JAK1, Janus kinase 1; AMPK, AMP-activated protein kinase.

**Figure 4 biomedicines-12-01525-f004:**
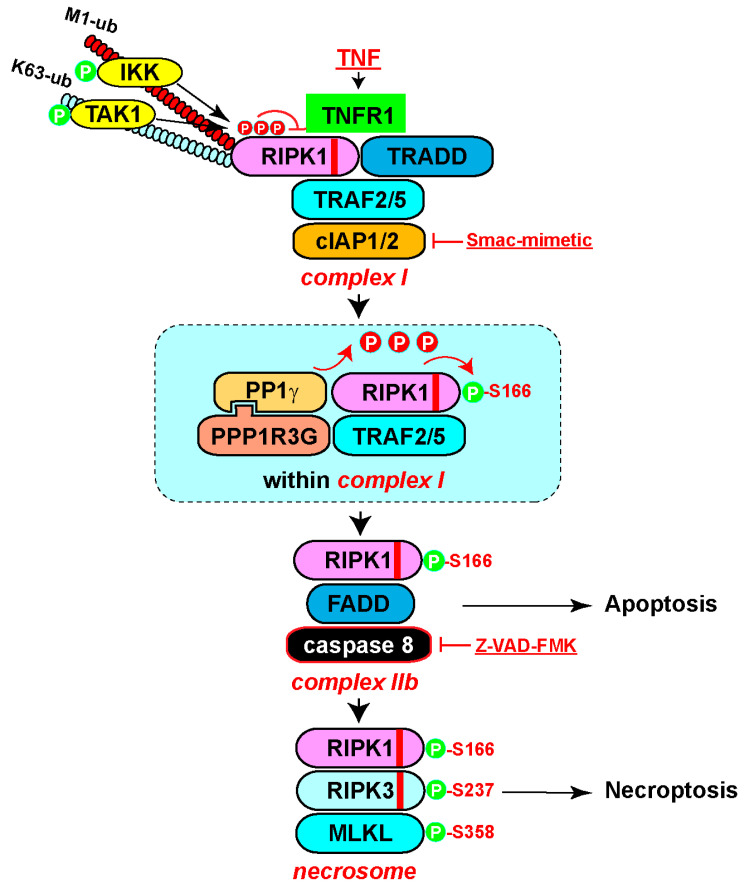
Regulation of RIPK1 activity through phosphorylation and dephosphorylation. This schematic illustrates the dynamic regulation of RIPK1-dependent cell death pathways. Inhibitory phosphorylation events on RIPK1, catalyzed by multiple kinases, serve as important checkpoints for cell death activation. Following cell death induction, the PPP1R3G/PP1γ holoenzyme is recruited to complex I to remove the inhibitory phosphorylation on RIPK1. This process enables RIPK1 autophosphorylation to activate its kinase activity. Consequently, activated RIPK1 triggers downstream signaling cascades, leading to apoptosis and necroptosis. The balance between inhibitory phosphorylation and PPP1R3G/PP1γ-mediated dephosphorylation serves as a key regulatory mechanism for RIPK1-dependent cell death processes. The red stripes in the diagram of RIPK1 and RIPK3 indicate the RHIM domain.

**Table 1 biomedicines-12-01525-t001:** RIPK1 inhibitors and clinical trials.

Drug (Company/Institution)	US Clinical Trial ID	Targeted Diseases	Phase
GSK2982772 (GSK, Brentford, UK)	NCT02903966	Ulcerative colitis	Phase IIa
	NCT02776033	Psoriasis	Phase IIa
	NCT04316585	Moderate to Severe Psoriasis	Phase I
	NCT02858492	Moderate to Severe Rheumatoid Arthristis (RA)	Phase IIa
GSK3145095 (GSK, Brentford, UK)	NCT03681951	Pancreatic ductal adenocarcinoma (PDAC)	Phase IIa-terminated
DNL104 (Denali, San Francisco, CA, USA)	NTR6257 (Netherlands)	Healthy Adults	Phase Ia-terminated
SAR443060 (DNL747) (Sanofi, Paris, France/Denali, San Francisco, CA, USA)	NCT03757351	Amyotrophic Lateral Sclerosis (ALS)	Phase Ib-terminated
	NCT03757325	Alzheimer’s disease (AD)	Phase Ib
SAR443122(DNL758) (Sanofi, Paris, France)	NCT04469621	Severe COVID-19	Phase Ib
	NCT05588843	Ulcerative colitis	Phase II
	NCT04781816	Cutaneous lupus erythematosus (CLEan)	Phase II
SAR443820 (DNL788) (Sanofi, Paris, France)	NCT04982991	Multiple Sclerosis Healthy Subjects	Phase I
	NCT05795907	ALS Healthy Volunteers	Phase I
	NCT05797701	ALS Healthy Volunteers	Phase I
	NCT05237284	ALS	Phase II-terminated
	NCT05630547	Multiple Sclerosis	Phase II
SAR443820 + Erythromycin/ Itraconazole (Sanofi, Paris, France)	NCT05797753	ALS Healthy Volunteers	Phase I
SIR1-365 (Sironax Beijing, China)	NCT04622332	Severe COVID-19	Phase I
R522 (Rigel, San Francisco, CA, USA/ Eli Lilly, Indianapolis, IN, USA)	[128]	Autoimmune and inflammatory diseases	Phase II
GFH312 (Genfleet, Shanghai, China)	NCT04676711	Healthy Adults	Phase I
	NCT05991362	Healthy Chinese Adults	Phase I
	NCT05618691	Peripheral Artery Disease (PAD)	Phase II-withdrawn
GDC-8264 (Genentech, San Francisco, CA, USA)	2019-002613-19 (Netherlands)	Healthy Adults	Phase I
	NCT05673876	Acute Graft-versus-host Disease	Phase Ib-terminated

Information for US clinical trials can be found at https://clinicaltrials.gov.

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
