# Peer review of "Regulation of RIPK1 Phosphorylation: Implications for Inflammation, Cell Death, and Therapeutic Interventions"

_biomedicines, 2024, doi:10.3390/biomedicines12071525_

Round 1

Reviewer 1 Report

Comments and Suggestions for Authors

The paper entitled Regulation of RIPK1 Phosphorylation: Implications for Inflammation, Cell Death, and Therapeutic Interventions brings important and up dated information about the importance of RIPK1 in inflammatory reaction regulation.

Observation

Abstract

Please mention the effects of RIPK1 phosphorylation on inflammatory process.

Introduction

It is improper to use the same sentence in Abstract and in the beginning of Introduction. Please reformulate it.

Please specify where the RIPK1 is located.

Please indicate the specific domains for apoptosis (death domain), necroptosis (kinase domain) and NF-kB activation (intermediate domain).

- line 78 - please change the last sentence, is confusing ; " activate inflammation to suppress cell death?"

Please make a scheme with the contribution of RIPK1 to the apoptosis and other forms of cells death  (necroptosis, piroptosis etc).

Therapeutic perspectives

Please specify if the treatment for RIPK1 inhibition is dose dependent or how to dose it to obtain a proper effect since this molecule can have beneficial and harmful effects. 

Author Response

The paper entitled Regulation of RIPK1 Phosphorylation: Implications for Inflammation, Cell Death, and Therapeutic Interventions brings important and up dated information about the importance of RIPK1 in inflammatory reaction regulation.

Abstract

Comment 1:Please mention the effects of RIPK1 phosphorylation on inflammatory process.

Response 1:We thank the reviewer for the suggestion. It is now stated in the abstract “Perturbations in RIPK1 phosphorylation status have profound implications for the development of severe inflammatory diseases in humans.”

Comment 2: Introduction

It is improper to use the same sentence in Abstract and in the beginning of Introduction. Please reformulate it.

Response 2: We thank the reviewer for the suggestion. The first sentence is now changed to “Receptor-interacting protein kinase 1 (RIPK1) is a crucial regulator of inflammation and cell death.”

Comment 3: Please specify where the RIPK1 is located.

Response 3: We now added the cellular location in the following sentence “The full-length human RIPK1 is a cytosolic protein consisting of 671 amino acids”.

Comment 4: Please indicate the specific domains for apoptosis (death domain), necroptosis (kinase domain) and NF-kB activation (intermediate domain).

Response 4: We sincerely thank the reviewer for this insightful suggestion. We appreciate the intention to provide readers with a clear understanding of RIPK1's structure-function relationships. However, as the reviewer correctly points out, the relationship between RIPK1's domains and their functions is not as straightforward as it might initially appear. While it's true that certain domains are primarily associated with specific functions (e.g., the death domain with apoptosis, the kinase domain with necroptosis, and the intermediate domain with NF-κB activation), the reality is more nuanced. For instance, as we discuss in our paper, the kinase activity of RIPK1 is crucial not only for necroptosis but can also play a role in apoptosis and pyroptosis under certain conditions.

Comment 5:  line 78 - please change the last sentence, is confusing ; " activate inflammation to suppress cell death?"

Response 5: We appreciate the reviewer for the suggestion. The sentence is now changed to “Both the MAPK pathway and the NFκB pathway activate gene expression that promotes inflammation and cell survival while suppressing cell death”.

Comment 6: Please make a scheme with the contribution of RIPK1 to the apoptosis and other forms of cells death  (necroptosis, piroptosis etc).

Response 6: We appreciate the reviewer's suggestion to create a scheme illustrating RIPK1's contribution to apoptosis and other forms of cell death. We would like to draw the reviewer's attention to Figure 2 in our manuscript, which we believe comprehensively addresses this request.

Comment 7: Therapeutic perspectives

Please specify if the treatment for RIPK1 inhibition is dose dependent or how to dose it to obtain a proper effect since this molecule can have beneficial and harmful effects. 

Response 7: We appreciate the reviewer's important question regarding the dose-dependency of RIPK1 inhibition treatment. We have addressed this concern by including a discussion of dose effects observed in phase I clinical trials of RIPK1 inhibitors, which states “Many of these RIPK1 inhibitors have successfully passed phase I safety tests, revealing important dose-dependent effects. For example, the brain-penetrant RIPK1 inhibitor DNL104 demonstrated a clear dose-response relationship in its safety profile. In the single-ascending dose group, DNL104 was well-tolerated across a range of doses, indicating a favorable safety profile at lower concentrations. However, in the multiple-ascending dose group, 37.5% of subjects experienced postdose liver toxicity, highlighting the potential for adverse effects at higher cumulative doses or with prolonged exposure[125]. This dose-dependent toxicity underscores the importance of careful dose optimization in RIPK1 inhibitor development. Some inhibitors have progressed to phase II trials, where dose-ranging studies are further refining the therapeutic window and optimal dosing regimens. However, as of yet, no RIPK1 inhibitors have advanced to phase III trials, partly due to the ongoing process of establishing the most effective and safe dosing strategies for long-term treatment”.

Reviewer 2 Report

Comments and Suggestions for Authors

The review is I think worthy for many scientists. The review describes the role of receptor-interacting protein kinase 1 (RIPK1) in the processes of apoptosis, necroptosis, and inflammation. It describes in very detail the activation of kinase by many other enzymatic processes, which is worthy and many valuable citations are there. There is also the way of kinase dephosphorylation. In these topics of phosphorylation and dephosphorylation in cell death, the paper focusses on the activity of the RIPK1 enzyme in vivo. Other authors also reviewed some parts of this review recently (Regulation of Inflammatory Cell Death by Phosphorylation).

Author Response

Comment 1: The review is I think worthy for many scientists. The review describes the role of receptor-interacting protein kinase 1 (RIPK1) in the processes of apoptosis, necroptosis, and inflammation. It describes in very detail the activation of kinase by many other enzymatic processes, which is worthy and many valuable citations are there. There is also the way of kinase dephosphorylation. In these topics of phosphorylation and dephosphorylation in cell death, the paper focusses on the activity of the RIPK1 enzyme in vivo. Other authors also reviewed some parts of this review recently (Regulation of Inflammatory Cell Death by Phosphorylation).

Response 1: We sincerely thank the reviewer for their thoughtful and encouraging comments on our review. We are pleased that the reviewer finds our work valuable for the scientific community, particularly in its detailed description of RIPK1's role in apoptosis, necroptosis, and inflammation. We appreciate the recognition of our comprehensive coverage of RIPK1 activation through various enzymatic processes and the inclusion of numerous valuable citations. We strived to provide a thorough and up-to-date analysis of the field, and we're glad this effort has been noticed. We acknowledge the reviewer's mention of other recent reviews on related topics, such as "Regulation of Inflammatory Cell Death by Phosphorylation." We are aware of the dynamic nature of this field and the contributions of our colleagues. While there may be some overlap in content, we believe our review offers a unique perspective and depth of analysis on RIPK1 phosphorylation and dephosphorylation specifically.

Reviewer 3 Report

Comments and Suggestions for Authors

In the manuscript, the authors described comprehensively the function and the regulation roles of receptor-interacting protein kinase 1 (RIPK1). More than one hundred papers were selected and reviewed to conclude the possible implications of RIPK1 for inflammation, cell Death, and therapeutic Interventions. In general, the study is interesting and the manuscript is well-organized and well-written. The study is of importance and may be interesting to a broad spectrum of readers in the field. The manuscript could be recommended for publication after some issues being addressing by the authors.

1. Whether the figures used by the authors requires permission is a concern.

2. A structure of RIPK1 reported could be added and that could help readers to have a structural understanding on this important protein.

3. Therapeutic perspectives section, line 424, some characters do not show properly.

4. For the Therapeutic perspectives section, the authors are highly recommended to add a table and summarize the clinical inhibitors of RIPK1 and also those are undergoing clinical trials at different phases. The information is useful and important for the researchers in the field, and may also increase the positive impact of the present study.

5. For the Future Perspectives section, it may need a bit more discussion, particularly which directions and what things could be suggested to encourage researchers for further investigations in the field in order to overcome the current challenges.

Author Response

In the manuscript, the authors described comprehensively the function and the regulation roles of receptor-interacting protein kinase 1 (RIPK1). More than one hundred papers were selected and reviewed to conclude the possible implications of RIPK1 for inflammation, cell Death, and therapeutic Interventions. In general, the study is interesting and the manuscript is well-organized and well-written. The study is of importance and may be interesting to a broad spectrum of readers in the field. The manuscript could be recommended for publication after some issues being addressing by the authors.

Comment 1. Whether the figures used by the authors requires permission is a concern.

Response 1: We appreciate the reviewer for raising this concern. All the figures were created by us and should not require permission from other journals.

Comment 2. A structure of RIPK1 reported could be added and that could help readers to have a structural understanding on this important protein.

Response 2: We appreciate the reviewer's suggestion to include a structural representation of RIPK1. While we agree that a 3D structure could provide valuable insights, we believe that the domain structure depicted in Figure 1 offers sufficient information for understanding the protein's functional organization in the context of this review. The domain diagram allows readers to easily correlate functional aspects discussed in the text with specific regions of the protein.

Comment 3. Therapeutic perspectives section, line 424, some characters do not show properly.

Response 3: We thank the reviewer for the keen observation. It is now fixed in the revised version.

Comment 4. For the Therapeutic perspectives section, the authors are highly recommended to add a table and summarize the clinical inhibitors of RIPK1 and also those are undergoing clinical trials at different phases. The information is useful and important for the researchers in the field, and may also increase the positive impact of the present study.

Response 4: We greatly appreciate the reviewer's valuable suggestion. In response, we have now included a comprehensive table summarizing the clinical inhibitors of RIPK1, including those currently undergoing clinical trials at various phases. This table has been added to the Therapeutic perspectives section. We agree that this information will be highly useful for researchers in the field and will enhance the impact of our study. The table provides a clear overview of the current state of RIPK1-targeted therapeutics, allowing readers to quickly grasp the progress in this area and identify potential areas for future research and clinical development.

Comment 5. For the Future Perspectives section, it may need a bit more discussion, particularly which directions and what things could be suggested to encourage researchers for further investigations in the field in order to overcome the current challenges.

Response 5: We sincerely appreciate the reviewer's valuable suggestion to enhance the Future Perspectives section. In response, we have significantly expanded this section to provide a more comprehensive discussion of future research directions and potential strategies to address current challenges in the field.

Round 2

Reviewer 3 Report

Comments and Suggestions for Authors

The authors have revised the manuscript. Some issues are still found in the revised version.

1. Figure 1 does not give any structural information about RIPK1. The authors please include a 3D structural presentation for it. That is available from PDB. A proper description is needed with suitable reference and PDB number cited.

2. I did not find any table added in the revised manuscript. Please double-checked.

Author Response

Dear Editor,

We appreciate the opportunity to submit a revised version of our manuscript "Regulation of RIPK1 Phosphorylation: Implications for Inflammation, Cell Death, and Therapeutic Interventions" (biomedicines-3029796). We have carefully addressed the feedback provided by the reviewers and the editor, incorporating their insights to strengthen our work. We are now delighted to submit the improved manuscript for your consideration.

We deeply appreciate the constructive feedback offered by reviewers. The following is our point-to-point response.

Reviewer 1:

The paper entitled Regulation of RIPK1 Phosphorylation: Implications for Inflammation, Cell Death, and Therapeutic Interventions brings important and up dated information about the importance of RIPK1 in inflammatory reaction regulation.

Abstract

Comment 1:Please mention the effects of RIPK1 phosphorylation on inflammatory process.

Response 1:We thank the reviewer for the suggestion. It is now stated in the abstract “Perturbations in RIPK1 phosphorylation status have profound implications for the development of severe inflammatory diseases in humans.”

Comment 2: Introduction

It is improper to use the same sentence in Abstract and in the beginning of Introduction. Please reformulate it.

Response 2: We thank the reviewer for the suggestion. The first sentence is now changed to “Receptor-interacting protein kinase 1 (RIPK1) is a crucial regulator of inflammation and cell death.”

Comment 3: Please specify where the RIPK1 is located.

Response 3: We now added the cellular location in the following sentence “The full-length human RIPK1 is a cytosolic protein consisting of 671 amino acids”.

Comment 4: Please indicate the specific domains for apoptosis (death domain), necroptosis (kinase domain) and NF-kB activation (intermediate domain).

Response 4: We sincerely thank the reviewer for this insightful suggestion. We appreciate the intention to provide readers with a clear understanding of RIPK1's structure-function relationships. However, as the reviewer correctly points out, the relationship between RIPK1's domains and their functions is not as straightforward as it might initially appear. While it's true that certain domains are primarily associated with specific functions (e.g., the death domain with apoptosis, the kinase domain with necroptosis, and the intermediate domain with NF-κB activation), the reality is more nuanced. For instance, as we discuss in our paper, the kinase activity of RIPK1 is crucial not only for necroptosis but can also play a role in apoptosis and pyroptosis under certain conditions.

Comment 5:  line 78 - please change the last sentence, is confusing ; " activate inflammation to suppress cell death?"

Response 5: We appreciate the reviewer for the suggestion. The sentence is now changed to “Both the MAPK pathway and the NFκB pathway activate gene expression that promotes inflammation and cell survival while suppressing cell death”.

Comment 6: Please make a scheme with the contribution of RIPK1 to the apoptosis and other forms of cells death  (necroptosis, piroptosis etc).

Response 6: We appreciate the reviewer's suggestion to create a scheme illustrating RIPK1's contribution to apoptosis and other forms of cell death. We would like to draw the reviewer's attention to Figure 2 in our manuscript, which we believe comprehensively addresses this request.

Comment 7: Therapeutic perspectives

Please specify if the treatment for RIPK1 inhibition is dose dependent or how to dose it to obtain a proper effect since this molecule can have beneficial and harmful effects. 

Response 7: We appreciate the reviewer's important question regarding the dose-dependency of RIPK1 inhibition treatment. We have addressed this concern by including a discussion of dose effects observed in phase I clinical trials of RIPK1 inhibitors, which states “Many of these RIPK1 inhibitors have successfully passed phase I safety tests, revealing important dose-dependent effects. For example, the brain-penetrant RIPK1 inhibitor DNL104 demonstrated a clear dose-response relationship in its safety profile. In the single-ascending dose group, DNL104 was well-tolerated across a range of doses, indicating a favorable safety profile at lower concentrations. However, in the multiple-ascending dose group, 37.5% of subjects experienced postdose liver toxicity, highlighting the potential for adverse effects at higher cumulative doses or with prolonged exposure[125]. This dose-dependent toxicity underscores the importance of careful dose optimization in RIPK1 inhibitor development. Some inhibitors have progressed to phase II trials, where dose-ranging studies are further refining the therapeutic window and optimal dosing regimens. However, as of yet, no RIPK1 inhibitors have advanced to phase III trials, partly due to the ongoing process of establishing the most effective and safe dosing strategies for long-term treatment”.

Reviewer 2:

Comment 1: The review is I think worthy for many scientists. The review describes the role of receptor-interacting protein kinase 1 (RIPK1) in the processes of apoptosis, necroptosis, and inflammation. It describes in very detail the activation of kinase by many other enzymatic processes, which is worthy and many valuable citations are there. There is also the way of kinase dephosphorylation. In these topics of phosphorylation and dephosphorylation in cell death, the paper focusses on the activity of the RIPK1 enzyme in vivo. Other authors also reviewed some parts of this review recently (Regulation of Inflammatory Cell Death by Phosphorylation).

Response 1: We sincerely thank the reviewer for their thoughtful and encouraging comments on our review. We are pleased that the reviewer finds our work valuable for the scientific community, particularly in its detailed description of RIPK1's role in apoptosis, necroptosis, and inflammation. We appreciate the recognition of our comprehensive coverage of RIPK1 activation through various enzymatic processes and the inclusion of numerous valuable citations. We strived to provide a thorough and up-to-date analysis of the field, and we're glad this effort has been noticed. We acknowledge the reviewer's mention of other recent reviews on related topics, such as "Regulation of Inflammatory Cell Death by Phosphorylation." We are aware of the dynamic nature of this field and the contributions of our colleagues. While there may be some overlap in content, we believe our review offers a unique perspective and depth of analysis on RIPK1 phosphorylation and dephosphorylation specifically.

Reviewer 3:

In the manuscript, the authors described comprehensively the function and the regulation roles of receptor-interacting protein kinase 1 (RIPK1). More than one hundred papers were selected and reviewed to conclude the possible implications of RIPK1 for inflammation, cell Death, and therapeutic Interventions. In general, the study is interesting and the manuscript is well-organized and well-written. The study is of importance and may be interesting to a broad spectrum of readers in the field. The manuscript could be recommended for publication after some issues being addressing by the authors.

Comment 1. Whether the figures used by the authors requires permission is a concern.

Response 1: We appreciate the reviewer for raising this concern. All the figures were created by us and should not require permission from other journals.

Comment 2. A structure of RIPK1 reported could be added and that could help readers to have a structural understanding on this important protein.

Response 2: We appreciate the reviewer's suggestion to include a structural representation of RIPK1. While we agree that a 3D structure could provide valuable insights, we believe that the domain structure depicted in Figure 1 offers sufficient information for understanding the protein's functional organization in the context of this review. The domain diagram allows readers to easily correlate functional aspects discussed in the text with specific regions of the protein.

Comment 3. Therapeutic perspectives section, line 424, some characters do not show properly.

Response 3: We thank the reviewer for the keen observation. It is now fixed in the revised version.

Comment 4. For the Therapeutic perspectives section, the authors are highly recommended to add a table and summarize the clinical inhibitors of RIPK1 and also those are undergoing clinical trials at different phases. The information is useful and important for the researchers in the field, and may also increase the positive impact of the present study.

Response 4: We greatly appreciate the reviewer's valuable suggestion. In response, we have now included a comprehensive table summarizing the clinical inhibitors of RIPK1, including those currently undergoing clinical trials at various phases. This table has been added to the Therapeutic perspectives section. We agree that this information will be highly useful for researchers in the field and will enhance the impact of our study. The table provides a clear overview of the current state of RIPK1-targeted therapeutics, allowing readers to quickly grasp the progress in this area and identify potential areas for future research and clinical development.

Comment 5. For the Future Perspectives section, it may need a bit more discussion, particularly which directions and what things could be suggested to encourage researchers for further investigations in the field in order to overcome the current challenges.

Response 5: We sincerely appreciate the reviewer's valuable suggestion to enhance the Future Perspectives section. In response, we have significantly expanded this section to provide a more comprehensive discussion of future research directions and potential strategies to address current challenges in the field.

We believe this revised version thoroughly addresses the reviewers’ comments and significantly enhances the quality of our manuscript. We look forward to your decision regarding its publication in Biomedicines.

Round 3

Reviewer 3 Report

Comments and Suggestions for Authors

The manuscript now could be recommended for publication.